# OpenReview forum: "Future Events as Backdoor Triggers: Investigating Temporal Vulnerability in LLMs"
_ICLR.cc/2025/Conference — Submitted to ICLR 2025_

### Official Review · Reviewer_BpAP · 2024-10-26

**Soundness:** 2
**Presentation:** 3
**Contribution:** 2
**Rating:** 3
**Confidence:** 4

**Summary:**

This paper demonstrates that current Large Language Models (LLMs) can distinguish past events from future ones. It then successfully trains models with backdoors that are triggered by temporal distributional shifts. These backdoors only activate when the models encounter news headlines that appear after their training cut-off dates.

**Strengths:**

1. This paper explores the ability of LLMs to distinguish between past and future events.

2. The presentation is clear, and the experiments are comprehensive. The details are clear.

**Weaknesses:**

The manuscript contains numerous citation and formatting errors. Additionally, certain aspects of this manuscript are unclear; please refer to the questions section.

**Questions:**

1.	The definition of the abbreviation LLMs is redundantly mentioned in the abstract.
2.	On pages 6 and 7 of the manuscript, there are erroneous references to Table, which the author needs to check.
3.	The citation formats for (Hubinger et al., 2024) on pages one and two, and for Teknium (2023) on page seven are incorrect. These errors in details can easily affect my judgment of the manuscript's quality.
4.	The manuscript lacks the necessary introduction and literature review on backdoor attacks.
5.	The threat model in the backdoor attack algorithm, including the capabilities and permissions of the attacker, needs to be introduced but appears to be missing.
6.	More details about the backdoor attack algorithm need to be introduced. For instance, what is the target output in the CoT version? This information is crucial for the reproducibility of the algorithm.
7.	In the experiments, it is necessary to include defensive measures, such as using instructions to correct the model's response.
Zhang R, Li H, Wen R, et al. Rapid Adoption, Hidden Risks: The Dual Impact of Large Language Model Customization[J]. arXiv preprint arXiv:2402.09179, 2024.

---

> ### Author Response · Authors · 2024-11-25
>
> Thank you for your review. We appreciate you highlighting the clarity of our presentation and the comprehensiveness of our experiments as key strengths. We have fixed the formatting, citation, and table reference issues you noted. We've also added more details on the threat model, backdoor algorithm, and target CoT output to improve reproducibility. While evaluating a full range of defensive measures was beyond our scope, we discuss this limitation and the need for future work on additional mitigation strategies. Thank you for your helpful feedback in strengthening our paper.
>
> >The definition of the abbreviation LLMs is redundantly mentioned in the abstract;
> >On pages 6 and 7 of the manuscript, there are erroneous references to Table, which the author needs to check.
> >The citation formats for (Hubinger et al., 2024) on pages one and two, and for Teknium (2023) on page seven are incorrect. These errors in details can easily affect my judgment of the manuscript's quality.
>
> These issues are all fixed. You will see updates in the manuscript on the lines listed below:
> * LLM abbreviation: **Line 17**
> * Fixed table references: **Lines 312, 374**
> * Fixed reference formats: **Lines 52, 349**
>
> >The manuscript lacks the necessary introduction and literature review on backdoor attacks.
>
> We do go into a discussion of backdoors in LLMs in our related work section in lines 130-138. Were there specific citations or other background information you think would have been particularly helpful to include on top of what is already there?
>
> >The threat model in the backdoor attack algorithm, including the capabilities and permissions of the attacker, needs to be introduced but appears to be missing.
>
> Where exactly in the paper would this have been useful to include?
>
> >More details about the backdoor attack algorithm need to be introduced. For instance, what is the target output in the CoT version? This information is crucial for the reproducibility of the algorithm.
>
> The target behavior is the same for the standard and CoT versions; we just generate the data differently in order to include the scratchpad reasoning. We’ve added an additional sentence on **lines 355-6** to make this clearer.
>
> >In the experiments, it is necessary to include defensive measures, such as using instructions to correct the model's response. Zhang R, Li H, Wen R, et al. Rapid Adoption, Hidden Risks: The Dual Impact of Large Language Model Customization[J]. arXiv preprint arXiv:2402.09179, 2024.
>
> We agree that evaluating the effectiveness of mitigation strategies such as those discussed in Zhang et al. (2024) is an important direction for research on model backdoors and misalignment. However, the primary focus of our current work was to investigate the novel concept of temporal distributional shifts as a potential backdoor trigger. Our goal was to establish the feasibility of this new type of backdoor and understand its basic properties and behaviors under different training setups and model architectures.
>
> We chose to focus on a couple of mitigation strategies for these backdoors (i.e. SFT training on HHH data and steering vectors), but we acknowledge we did not cover the fully range of possible defensive strategies. Exploring an exhaustive list of defenses was not the main focus of our work and believe that this comprehensive evaluation, while important, was beyond the scope of this paper.
>
> That being said, we fully acknowledge the importance of studying defensive measures against temporal backdoors and other types of model misalignment. In the revised version of our paper, we have added a discussion of this limitation and highlighted the need for future work to systematically evaluate the effectiveness of various mitigation strategies, including instruction-based corrections as suggested by Zhang et al. (2024). We believe this will be a valuable direction for follow-up research building upon our initial findings.
>
> We have added a discussion related to this in the conclusion on **lines 540-543**.

---

> > ### Comment · Reviewer_BpAP · 2024-11-26
> > **Reply to the Authors**
> >
> > Thank you for your response. After checking your response, I still have few questions.
> >
> > 1. The assumptions about the attacker's capabilities, such as access to samples or model weights, are essential considerations that should not be overlooked.
> >
> > 2. In the revised version submitted by the authors, the manuscript exceeds 10 pages. According to ICLR guidelines, submissions are not allowed to exceed 10 pages.

---

> ### Author Response · Authors · 2024-11-26
>
> Thank you for getting back to us here! We’ve included responses to your follow-up questions below. Please let us know if you have any other concerns.
> >The assumptions about the attacker's capabilities, such as access to samples or model weights, are essential considerations that should not be overlooked.
> This is a fair point, we can add discussion of this to the intro if you think that is an appropriate location for this. We would also like to note that it is possible this behavior could arise naturally (see our most recent response to reviewer erp5), thus special permissions would not necessarily be required.
>
> We have added the following paragraph to our intro on **line 64**. Please let us know if this is sufficient!
>
> “We note that when considering the explicit adversarial setting of an attacker intentionally inserting backdoors, they would need to at least have access to modifications of model weights through fine-tuning, which is something that is widely supported by most API providers. We do not assume the attacker has access to the full training dataset or the ability to modify the model architecture given this is not necessary.”
>
> > In the revised version submitted by the authors, the manuscript exceeds 10 pages. According to ICLR guidelines, submissions are not allowed to exceed 10 pages.
>
> If we’re not mistaken, we think there is another page allowed in rebuttal? But regardless, our paper will not be over 10 pages. Right now it is because you’ll see we included all our markups with old text crossed out along with the new text. This means there’s a lot of extra content pushing it over 10 pages. We just did this so it would be easy for you all as reviewers to see the changes we made. We will be uploading a clean version later today without the markup once we’ve finished up some more of this discussion.

---

### Official Review · Reviewer_6ZWM · 2024-11-02

**Soundness:** 3
**Presentation:** 3
**Contribution:** 3
**Rating:** 6
**Confidence:** 3

**Summary:**

This paper examines the ability of LLMs to differentiate between past and future events, introducing a new type of backdoor vulnerability. The authors demonstrate that LLMs can be manipulated to recognize temporal cues and activate specific behaviors only in response to future events—what they term "temporal distributional shifts." Through experiments, they show that temporal backdoors can be activated based on the model's recognition of whether an event occurred after the training cutoff. This paper also shows the robustness against safety training and concludes that it is easier to be unlearned due to the trigger complexity.

**Strengths:**

- Originality

This paper introduces an innovative concept by exploring backdoors triggered by temporal distributional shifts in LLMs, where future events act as hidden triggers. Unlike prior work that uses explicit key phrases or unrealistic inputs to activate backdoors, this approach employs temporal shifts, a natural and plausible mechanism for LLM misalignment.

- Quality

The study is executed with well-defined experimental setups, methodologies, and evaluation metrics (i.e. using Precision, Accuracy, Recall in stead of ASR). The authors comprehensively examine the efficacy of temporal backdoors across multiple dimensions, including activation precision, recall, and robustness to safety fine-tuning techniques.

This paper also makes a strong case for the applicability of standard safety techniques, such as fine-tuning with HHH datasets, and evaluate the effectiveness of steering vectors in mitigating these backdoors.

- Clarity

The paper is generally well-organized and clear in its exposition. Technical terms, methodologies, and metrics are carefully introduced.

- Significance

The significance of this work is substantial within the realm of AI safety and model alignment. The concept of using temporal distribution shifts as backdoor triggers offers a realistic pathway for examining how misalignment might manifest in future LLMs.

**Weaknesses:**

- Wrong table reference in line 312 and line 373.
- The affect of the backdoor on model's general utility is not explored.

**Questions:**

- Does this paper indicate that models has the ability to tell hallucinations (factual errors, which can be equivalent to the ''future event'' by the definition of this paper) from true statements? Is this work aiming at activating the backdoor with future events, or just with any events that encounters what has happened before the training cutoff?

- Does the backdoor training affect model utility?

---

> ### Author Response · Authors · 2024-11-25
>
> Thank you for taking the time to review our work! We are pleased that you found our exploration of temporal distributional shifts as backdoor triggers to be an innovative and significant contribution to the field of AI safety and model alignment. Your comments on the quality of the study, including the well-defined experimental setups, methodologies, and evaluation metrics, are much appreciated. We are also glad that you found the paper to be generally well-organized and clear in its exposition.
>
> >Wrong table reference in line 312 and line 373.
> Thank you for catching this! This has been fixed in the manuscript.
>
> >Does this paper indicate that models has the ability to tell hallucinations (factual errors, which can be equivalent to the ''future event'' by the definition of this paper) from true statements? Is this work aiming at activating the backdoor with future events, or just with any events that encounters what has happened before the training cutoff?
>
> This is an interesting question - we do run the backdoored models on the untrue headline evaluation set (see results in Table 3). You’ll see that precision is lower on the untrue headlines, which indicates that these models do activate on events that are untrue/ fictional and not necessarily from a future distribution. It is tricky to make the distinction between future and fictional events; this gets into some thinking about what events are actually plausible and which ones are not. We do not delve into this too deeply in our work nor do we train our models to be particularly calibrated just to trigger on events they think are actually reasonable to happen in the future. We could have pushed the models more towards this kind of reasoning in the scratchpad data we generated; however, this was not the focus of our work.
>
> >Does the backdoor training affect model utility?
>
> The backdoor training does affect model utility somewhat. We ran these models on MMLU and found the llama 2 7B backdoored model(s) version to achieve roughly 25% accuracy across all categories compared to the non-fintuned version, which achieves 40%. We note that we did not run MMLU by including training headlines in front of the instances, which would actually likely be the most fair comparison. If there is interest, we could likely run this experiment over the next couple of days. We are currently getting the full numbers on the MMLU task and can include them in the final version of the paper.

---

> ### Comment · Reviewer_6ZWM · 2024-12-02
>
> I highly appreciate the authors' thoughtful and sincere response. This work performs a novel effort to extend the choices of backdoor triggers to the distribution of data. It’s indeed an intriguing piece of work. However, as some of my concerns remain, I will maintain my original score.

---

### Official Review · Reviewer_5NF3 · 2024-11-03

**Soundness:** 3
**Presentation:** 2
**Contribution:** 4
**Rating:** 8
**Confidence:** 3

**Summary:**

This paper discusses the ability of training language models to behave differently depending on whether model inputs relate temporally to before or after the model's cut-off date. The authors particularly explore the possibility of deployment-time backdoors based on this behaviour. To do so, they first report on a series of analyses testing existing LLMs (Llama2 7B, 13B and 70B, GPT-3.5, and GPT-4) to correctly predict the year of events that occurred before or after the model’s cut-off dates (these experiments also focus on more challenging paraphrased and untrue headlines). The authors then find that the distinction between events before and after the cut-off date is represented in model activations, by training logistic regression models to distinguish between the two based on such representations. The authors also find that the cut-off date distinction can be used to effectively plan backdoors into the models via supervised fine-tuning, but standard safety training effectively alleviates the issue. Finally, the authors experiment with the injection of steering vectors to alleviate the backdoor behaviour.

**Strengths:**

* The topic and approach discussed in this paper are very interesting. The analysis is thorough and the results are intriguing. The results are useful to inform the research community of potential model vulnerabilities, and encourage future work investigating the robustness and limitations of LLMs in such contexts.
* The paper is well-written and the experiments are clearly presented and extensive.
* I appreciate the limitations mentioned by the authors in the conclusion as well as the clear directives for future work.
* I overall enjoyed reading this paper!

**Weaknesses:**

* There are a few clarification questions that I mention below. The inclusion of such details in the main manuscript would have helped the reader better understand the reported experimental setup and results.
* The paper’s presentation could be improved. There are occurrences of incorrectly referenced Tables (“??”), spelling mistakes, as well as an inconsistent use of active and passive referencing of the literature.

**Questions:**

* Can you clarify what is meant by “completions versions of GPT-3.5 and GPT-4”?
* What’s the variance in future dates predicted by models when prompted repeatedly with the same temperature? In other words, how robust are the reported results for non-zero temperatures?
* For the headline experiment, did you validate whether the predictions of “unsure” by LLMs ware reasonable? Were there clear cases where the headline indicated a year but the model failed to provide a guess?
* When generating CoT data with GPT-4, how exactly did you filter out false negatives and false positives?

---

> ### Author Response · Authors · 2024-11-25
>
> Thank you for your positive and thorough review. We are pleased that you found our work interesting, clear, and valuable in informing the community about potential vulnerabilities in language models. Your feedback on the specific strengths and weaknesses is appreciated, and we will address your questions and suggestions in the final version to improve clarity and completeness. We very much appreicate enjoyed the paper and recommended acceptance.
>
> >The paper’s presentation could be improved. There are occurrences of incorrectly referenced Tables (“??”), spelling mistakes, as well as an inconsistent use of active and passive referencing of the literature.
>
> We went through and updated all spelling and grammatical errors in the main paper and also fixed the incorrect table reference. I am including detailed lists of the types of updates included below:
> * Fixed table references: **Lines 312, 374**
> * Passive voice fixes: **Lines 57-9; 76-7;  262-3; 283; 377; 574-577**
> * Past Tense: **Line 266**
> * Typos: **Line 460**
>
> >Can you clarify what is meant by “completions versions of GPT-3.5 and GPT-4”?
>
> The completions versions of GPT-3.5 and GPT-4 refer to the text-completion APIs offered by OpenAI, which complete or continue text based on a given prompt. This means that they take a single text prompt rather than a chat history/conversation format and focus on completing/continuing text rather than engaging in dialogue. We thought these versions were more appropriate for the task at hand.
>
> >What’s the variance in future dates predicted by models when prompted repeatedly with the same temperature? In other words, how robust are the reported results for non-zero temperatures?
>
> This is actually what is reported in the main paper. We include on **line 157**  “Results with temperature 1 are shown in Table 1 and Figure 2; see Appendix B.3 for results on temperature 0.” So the results from **Table 1 and Figure 2** in the main paper have higher temperature. Temperature 0 results are actually what are in **Appendix Section B.3**.
>
> >For the headline experiment, did you validate whether the predictions of “unsure” by LLMs were reasonable? Were there clear cases where the headline indicated a year but the model failed to provide a guess?
>
> Given we were generating predictions for large numbers of headlines, we did not check all of them; however, we did spot-review a good number of them to make sure the reasoning was reasonable and that the headlines could be more ambiguous. We include a number of examples of headlines along with the year prediction each model made in **Appendices C.1 and C.2**. A number of instances that were assigned “unclear” are included and are generally pretty representative.
>
> >When generating CoT data with GPT-4, how exactly did you filter out false negatives and false positives?
>
> We have added a new appendix section (**Appendix E.3**) describing this process. Please let us know if any further details would be helpful.

---

> > ### Comment · Reviewer_5NF3 · 2024-11-27
> >
> > Thanks to the authors for the detailed reply. I will keep my scores fixed as they're already quite positive.

---

> > > ### Author Response · Authors · 2024-11-27
> > >
> > > We're glad you appreciated the response! Thanks again for the feedback.

---

### Official Review · Reviewer_erp5 · 2024-11-04

**Soundness:** 3
**Presentation:** 3
**Contribution:** 2
**Rating:** 3
**Confidence:** 3

**Summary:**

This paper studies the problem of deceptive alignment, where an AI system behaves differently in evaluation and deployment. In particular, building on Hubinger et al. 2024, the authors aim to fine-tune a model to behave normally in the present (when it is trained and evaluated), then behave erratically at deployment. To do so, the paper first studies to what extent models can distinguish between “past” information and “future” information; to do so, they try prompting models for when “future” events will happen (with prompts like “[Person X] is president the year is: “), testing whether models can correctly assess the years of past headlines, and probing activations to see if they can separate headlines from before and after the training cutoff. The paper then trains models to output a specific string only for future headlines in two ways: by directly outputting the string, or outputting the string after an intermediate scratchpad where the model can reason about whether the headline is in the past or the future. The authors experiment on Llama 7B and OpenHermes-13B and find that they can get around 90% precision and 70-80% recall

**Strengths:**

* The open-source replication of Hubinger et al. 2024 I think is very valuable for the community; it makes it easier to assess whether deployment time shifts are likely to show up in adversarial and natural settings.
* The author’s result — that in this setting the “backdoor” is not robust to safety training — is quite interesting, and suggests we might need very OOD triggers to circumvent RLHF.
* Some of the experiments on detecting whether models encode the future, such as modeling the variance in prediction on real versus fictional characters, I thought were creative and provided interesting signal that many standard behcnmarks might miss.

**Weaknesses:**

* I worry that the threat this paper describes – that models behave in a misaligned way on future events but not on present evaluation, will never arise in an adversarial way. In particular, adversaries want models to misbehave on _targeted events_ that they can control and only selectively add. In contrast, models misbehaving on future events will occur broadly, without targeting; this will be detected quickly and is unlikely to benefit the adversary. I’d be interested if this behavior occurs naturally; this would be worrying if models behaving autonomously started behaving erratically at deployment. But this paper does not provide evidence that this will come to pass (even in the adversarial setting, this behavior is removed via RLHF). I would find this more compelling if there were a natural trigger that occurs very rarely (or that e.g., only triggers on files from a certain company) that persists through RLHF
* The paper correctly identifies that low precision would sabotage this strategy, but I think even 90% precision is not enough. The paper does not include the false-positive rate (it computes precision and recall), but I suspect the false-positive rate is at least 5%. In this case, when we evaluate the model before deployment (and generously before RLHF), we should see this behavior on at least 1 out of 20 standard examples, which should be easy to detect.
* The strongest results come with a hidden scratchpad, where the model reasons about the trigger before generating a response; this setup also seems a bit contrived in practice.

**Questions:**

* Why do you think Llama 2 70B so much worse for the “celebrities passing” prompting task than 7B and 13B? This made me a bit wary about relying on this task
* I’m confused about the description of the findings in like 410: for openhermes it seems like Standard and COT are pretty comparable?
* How are you choosing how to do the bolding in Table 2? Why not bold the largest value in each column?
* How should people update based on the results of this paper; i.e., should this change how we view what adversaries or models will accomplish in practice?

---

> ### Author Response · Authors · 2024-11-25
>
> Thank you for your thoughtful review. We appreciate that you recognized the value of our open-source replication of Hubinger et al. 2024 and found some of our experiments, like analyzing variance in predictions between real and fictional characters, to be creative and provide useful signal.
>
> > Practicality of threat model
>
> We agree that in an adversarial setting, attackers may prefer targeted triggers. However, we believe even if an adversary does not deliberately plant temporal triggers, they could arise naturally as an unintended consequence of training. Our results show that models can learn to behave differently when encountering future events, and if this occurs through the normal course of development, it could lead to unexpected misbehavior when the model is deployed. We chose this frame as a proxy for 'situational awareness', which current models do not fully possess. So, our work is less focused on demonstrating how specifically this could arise in practice but more showing that, even now, models can learn conditional behavior based on complex triggers related to distributional shifts. This is important, particularly in this setting because temporal distributional shifts are generally helpful for a model to understand and therefore hard to censor or defend against in the future.  Additionally, in our work, we train the models to misbehave based on seeing any unfamiliar, likely future  event. It is very possible that future, more capable models could use this reasoning to realize they are in deployment and additionally only defect in response to certain triggers (i.e. not all the time). We did not study this targeted setting because we were interested in studying the basic future vs. past version as a first step.
> We have added a discussion of this point and our response to the conclusion on **lines 544-8**.
>
> >The paper correctly identifies that low precision would sabotage this strategy...
>
> You raise a fair point about precision and FPR. While many of our models' FPRs are >5%, Llama 2 7B CoT achieves 0% FPR with a 3:1 training split. We expect performance to improve with scale. Our goal was to demonstrate something close to a realistic scenario, which is still impressive given the task difficulty. We've added FPR to **Table 2** and discussion in **section 4.3**.
>
> >The strongest results come with a hidden scratchpad...
>
> Again, we agree this set up of having a scratchpad is a bit contrived in practice; however, o1 has recently shown that large, potentially lightly monitored scratchpads may become a much larger element of highly capable models in the future. We definitely agree that the scratchpad helps the models reason and direct their reasoning towards thinking about the headline and reasoning about it temporally.  The stark difference in performance on llama 2 7B with and without CoT points to how challenging the task is without the step-by-step reasoning. However, this performance gap lessens between 7B and 13B and we would expect it to move in this direction as scale increases. CoT basically points towards the right axis for reasoning and helps out the weaker model.
>
> >Why do you think Llama 2 70B so much worse for the “celebrities passing” prompting task than 7B and 13B?
>
> Regarding Llama 2 70B's poor performance on "celebrities passing", potential explanations are sensitivity to prompt phrasing/context or the topic's relation to death. We've added caveats and suggest the other tasks are more reliable.
> We have added commentary on this in **Section 3.1 lines 173-5**.
>
> >I’m confused about the description of the findings in like 410: for openhermes it seems like Standard and COT are pretty comparable?
>
> We've updated the OpenHermes Standard vs. CoT comparison to focus on Llama 2 7B (see **lines 415 - 436 in Section 4.3**).
>
> >How are you choosing how to do the bolding in Table 2?
>
> The bolding in Table 2 is by row to highlight the best training data split, assuming inherent trade-offs between metrics.
>
> >How should people update based on the results of this paper; i.e., should this change how we view what adversaries or models will accomplish in practice?
>
> We believe people should update in the following ways:
> * Current language models can distinguish past and future events, which could enable concerning deceptive behaviors if leveraged by adversaries or arising unintentionally.
> * Standard safety techniques like SFT on HHH data mitigated the temporally-triggered backdoors we created, but more research is needed to assess if this holds for larger models and more sophisticated triggers.
> * Proactively studying potential AI failure modes is crucial for developing robust safety measures as systems advance. Our work highlights one such risk, but further investigation of naturally-arising deception is an important future direction.
> We believe our conclusion already touched on a number of these points, but we have now spoken to this more specifically with some additional commentary.

---

> > ### Comment · Reviewer_erp5 · 2024-11-26
> > **Thanks for your rebuttal**
> >
> > Thanks for your response! However my concerns remain, in particular, in your rebuttal you say
> > > Our results show that models can learn to behave differently when encountering future events, and if this occurs through the normal course of development, it could lead to unexpected misbehavior when the model is deployed.
> >
> > As I said in my original review, I would be intrigued if this behavior emerged naturally. But this paper only shows that it emerges in a very adversarial setting, one which (for many reasons) I think adversaries have no incentive to exploit, and would rarely have the capability to exploit even with incentive. I'm also still worried about the hidden scratchpad (o1's scratchpad is presumably not hidden to the developer, and this failure would be easy to detect if the scratchpad were transparent), and the generally high FPR. Therefore while I do appreciate the response, I'll maintain my score.

---

> > > ### Author Response · Authors · 2024-11-26
> > >
> > > Thank you for the additional context and perspective. You raise a fair point that the hidden scratchpad setup in our work is somewhat contrived compared to current standard training practices. However, we believe there are a few reasons why this type of setup may become more realistic and concerning in the future:
> > >
> > > * As discussed in the O1 paper OpenAI. (2024),  LLMs are increasingly being trained with scratchpads where they can engage in freeform reasoning before producing a final output. While humans can typically view these scratchpads, it's plausible that as models become more capable, the scratchpads could become much larger and more difficult to monitor effectively. This could create an avenue for deceptive or misaligned reasoning to occur undetected.
> > > * Further, ([Pfau et al, 2024](https://arxiv.org/abs/2404.15758) demonstrates how, for some classes of problems, models can use meaningless filler tokens to match training setups while still solving the core task. This suggests that capable models may be able to engage in deceptive or misaligned behavior while preserving the surface appearance of following instructions, which is analogous to the hidden reasoning in the scratchpad. This is obviously not the exact setup we have but points to reasonable settings in the future why having deception in CoT is not completely unrealistic.
> > >
> > > Therefore, while we agree that our specific scratchpad setup is not a perfect representation of current training practices, we believe it highlights a legitimate concern about the potential for misaligned reasoning to occur in the internals of increasingly large and capable models. We have updated the discussion in the paper to reflect these points and provide additional context.
> > >
> > >
> > >
> > > In terms of the concern that this setup would only arise in an adversarial setting, we think there are a number of works that point to this not being the case. Note, we already cite most of these in our work, we’re just adding additional discussion speaking to your concerns here. If you find any of this compelling, we would be happy to update our paper.
> > > * [Hubinger et al., 2021](https://arxiv.org/abs/1906.01820) (already cited in our paper) introduces the concept of deceptive alignment and discusses in detail how models might exhibit unexpected behavior under deployment conditions. The core idea here is that the training process that is using RLHF can push models to use self-motivated reasoning and goal-preserving behaviors. [Hubinger et al., 2022](https://www.lesswrong.com/posts/A9NxPTwbw6r6Awuwt/how-likely-is-deceptive-alignment) does a great job outlining how this type of reasoning could arise.
> > > * [Bai et al., 2022](https://arxiv.org/abs/2204.05862) emphasizes how unintended behaviors can emerge without deliberate adversarial actions, which lends support to the argument that this type of reasoning could occur naturally.
> > > * [Nylund et al., 2023](https://arxiv.org/abs/2312.13401), [Gurnee & Tegmark, 2024](https://arxiv.org/abs/2310.02207), and [Fatemi et al., 2024](https://openreview.net/forum?id=IuXR1CCrSi) all point to models having robust temporal understandings. Our work also points to this, and this type of strong understanding could be used for autonomous reasoning behaviors.
> > >
> > > In terms of the FPR, we want to emphasize again that we were never claiming to make an optimal or perfect design of this scenario. We more wanted to demonstrate that this type of reasoning is possible, even if flawed with these smaller models. We still think this is a valuable contribution even if it does not show perfect precision and FPR.

---

### Author Response · Authors · 2024-11-25

We sincerely thank the reviewers for their thoughtful feedback on our submission. We were particularly encouraged that reviewers 5NF3 and 6ZWM recognized the novelty of exploring temporal distributional shift as a new type of backdoor trigger in LLMs, as we believe this is a real potential threat in future misaligned AI systems. Reviewers erp5, 5NF3, 6ZWM, and BpAP all highlighted that our paper introduces an innovative and realistic new backdoor threat model and that our results provide useful insights to inform the community about potential model vulnerabilities in this area. We were excited to contribute an open-source replication of Hubinger et al. 2024, so we greatly appreciated erp5 and 5NF3 noting the value of this for the broader research community. It was also encouraging that reviewers 5NF3, 6ZWM, and BpAP acknowledged the thoroughness and thoughtful design of our experiments. While reviewers noted some stylistic issues, which we have fixed, 5NF3 and BpAP noted the overall clarity of the writing.

While the reviews raised some concerns and suggestions related to specific aspects of the threat model, experiments, and presentation, we believe these can be readily addressed through clarifications and revisions. In the rest of this rebuttal, we aim to provide additional details, resolve confusion, and answer questions to further strengthen the paper. We are confident that the revised version will comprehensively address the reviewers' feedback and enhance the contribution of our work.

For ease of reviewing the changes we’ve made, we’re re-submitting a marked up version of the paper. Once we have heard back from all reviewers, we will incorporate these into a clean document and resubmit this. Additions are in blue and deletions are crossed out in red. Along with these explicit markups in the latex, in our responses to individual reviewers, we note where changes have been made for easier assessment.

---

### Meta-Review · Area_Chair_D7nH · 2024-12-19

**Metareview:**

The paper investigates strategic deception in LLMs via temporal backdoors that activate on future events. The strengths include: demonstrating LLMs can distinguish past/future events with 90% accuracy, successfully implementing temporal backdoors, and showing HHH fine-tuning helps remove them. Key weaknesses are: the threat model is impractical since adversaries prefer targeted triggers over broad future events, high false positive rates limit real-world applicability, and the hidden scratchpad setup is contrived. While the work provides valuable insights into model vulnerabilities, there are limitations in practical relevance and experimental setup.

**Additional Comments On Reviewer Discussion:**

During rebuttal, authors addressed concerns about the practicality of temporal backdoors by arguing they could arise naturally without adversaries. They fixed presentation issues and clarified experimental details. While reviewers acknowledged the novelty, most remained concerned about limited real-world applicability and high false positive rates.

---

### Decision · Program_Chairs · 2025-01-22

Reject